# Bulk locality and asymptotic causal diamonds

**Chethan Krishnan**[1][*]

**1** Center for High Energy Physics, Indian Institute of Science, Bangalore 560012, India

[*] chethan.krishnan@gmail.com

## Abstract

In AdS/CFT, the non-uniqueness of the reconstructed bulk from boundary subregions has motivated the notion of code subspaces. We present some closely related structures that arise in flat space. A useful organizing idea is that of an *asymptotic* causal diamond (ACD): a causal diamond attached to the conformal boundary of Minkowski space. The space of ACDs is defined by pairs of points, one each on the future and past null boundaries, $\mathcal{I}^{\pm}$. We observe that for flat space with an IR cut-off, this space (a) encodes a preferred class of boundary subregions, (b) is a plausible way to capture holographic data for local bulk reconstruction, (c) has a natural interpretation as the kinematic space for holography, (d) leads to a holographic entanglement entropy in flat space that matches previous definitions and satisfies strong sub-additivity, and, (e) has a bulk union/intersection structure isomorphic to the one that motivated the introduction of quantum error correction in AdS/CFT. By sliding the cut-off, we also note one substantive way in which flat space holography differs from that in AdS. Even though our discussion is centered around flat space (and AdS), we note that there are notions of ACDs in other spacetimes as well. They could provide a covariant way to abstractly characterize tensor sub-factors of Hilbert spaces of holographic theories.

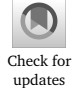
---

# 1 Introduction

We will start by observing a few reasons to believe that holography is likely to be a universal feature of quantum gravity[1].

Firstly, observables in a diffeomorphism invariant (quantum) theory with a dynamical metric, are naturally integrals over all of spacetime[2]. This means that bulk spacetime is a dummy variable in quantum gravity and that observables are supported only at the boundary. Secondly, black holes seem to be the highest entropy objects in theories of gravity, and their entropy scales with area and not volume. This hints at the fact that the Hilbert space size of a region of space in quantum gravity does not scale with volume[3]. Thirdly, the open-closed duality of string theory is a suggestion that strings that contain gravity are dual to strings that contain only non-gravitating fields. Since open strings are excitations of D-branes which wrap submanifolds, it is perhaps not too far-fetched (at least not in hidsight!) to think that there should be a desription for gravity in terms of a lower dimensional non-gravitataional theory.

In the above paragraph, we have not mentioned the word "AdS" even once. This is because we expect holography to hold more generally than in AdS – in fact, the idea of holography was introduced by 'tHooft and Susskind [1, 2] long before the AdS/CFT correspondence [3–5]. However there are two features that make holography shockingly more impressive in AdS than elsewhere. One is that we know explicit examples of holographic duals (eg. [3] and many more) in some asymptotically AdS spaces thanks to string theory[4]. Secondly, at least at the semi-classical level, in asymptotically AdS spaces we know how to formulate a correspondence between bulk calculations and boundary calculations [5]. This is in big parts due to the fact that the holographic boundary in AdS/CFT is as close to a *physical* boundary as one can hope for.

In some ways, these two aspects of AdS/CFT are quite independent. The former is a definition of the relevant theories that form the two ends of the duality, while the latter is a structural statement about how holography is implemented. In particular, the (semi-classical) statement of the correspondence only seems to require very coarse-grained features of these dual theories: the structure of the conformal boundary, the actions of the bulk isometries, and how classical fields propagate in the bulk[5]. This suggests the possibility that if we could guess the analogous structures in other spacetimes, we might be able to make progress with only minimal knowledge about the specific theories involved[6]. In this paper, we hope to take a very small step in this rough direction.

Our starting point is the observation that the natural "unit" of holography in a region of

---

[1]This is clearly a vague statement. The challenge is to clarify in what precise sense this is true in various theories/spacetimes/etc.

[2]The local metric acquires infrastructural meaning when $G_N \to 0$, where gravity is weak and close to non-dynamical. The key point is that in a spacetime with a non-dynamical metric, diffeomorphism invariance ends up becoming a trivial gauge redundancy: one can gauge-fix the metric to a form where its isometries (if any) are manifest, and then use coordinates in that gauge to label points in spacetime. This is what one does in Poincare invariant field theory, which has local observables. But when the metric is dynamical, it is not clear how to solve away the gauge redundancy, and we seem forced to go to the boundary of spacetime to find observables. Though perhaps not as widely appreciated as they should be, these facts are known one way or another since the days of Einstein's "hole argument".

[3]Dynamical diff invariance and black holes are two defining features of gravity, and the takeaway from the above discussion is that both of them ndicate a holographic definition of quantum gravity.

[4]See [6] for a non-AdS example for quantum gravty.

[5]Note that the existence of a sparse spectrum and a large central charge [7] in the CFT are *pre-conditions* for a semi-classical weakly curved description, and so are not structural to the semi-classical correspondence in the sense that we use here.

[6]Of course there exists the possibility that this might give us enough of a hint to characterize (perhaps even fully explicitly) holographic theories.

spacetime is a causal diamond[7]. The time evolution inside the causal diamond (at least at the semi-classical level) is entirely determined by the data on the diamond. Versions of this idea have been exploited in many interesting ways for almost two decades now, and many interesting papers have explored various aspects of it [8–10, 12–15]. We will add a new twist to this ingredient, by introducing the notion of an *asymptotic causal diamond*. In most of our discussion, we will stick to flat space for concreteness, but in the final section we will make some comments about other spacetimes.

The motivation for introducing an asymptotic causal diamond is *local* bulk reconstruction[8]. In AdS, we know that local bulk reconstruction can be accomplished from boundary *subregions* through causal wedges and related ideas [16–18], and we wish to know what are the structures required for accomplishing this more generally, in particular in flat space. It is easy enough to convince oneself that in flat space, the plausible answer is a causal diamond attached to the conformal boundary. This is equivalent to choosing two points, one each on the two null boundaries. It naturally encodes a boundary subregion (in a manner we will make precise), while at the same time we expect it to contain precisely enough data to reconstruct local bulk regions. The idea of taking asymptotic causal diaomnds seriously for holographic purposes beyond AdS/CFT is likely to be fruitful: we will present developments along multiple directions using these ideas in some follow up papers [19, 20], but will limit ourselves in this paper to making clear that the local bulk structure that arises from them in flat space is isomorphic in many ways to that which arises in AdS holography. To do this, simple geometric arguments will suffice.

We will phrase things in the language of the quantum error correction idea [21–23] that has recently become prominent in the AdS/CFT context. The key point here is that to make sense of local bulk observables in AdS/CFT, when there are multiple regions from which one can re-construct the "same" bulk point, we need to think of the bulk spacetime as a code subspace of the quantum gravity Hilbert space. The main motivation for this argument, as presented in [21], is that the bulk regions reconstructed from boundary subregions has a certain natural union and intersection structure. We will demonstrate in this paper that this structure arises naturally in our asymptotic causal diamond reconstruction as well. An IR cut-off plays a role as well, in making the picture sensible: this should be viewed as the requirement that we have to isolate the gravitating system in some suitable sense before we can describe it holographically.

This paper is a small deformation of many pre-existing ideas, we will mention two that we feel are especially significant. The idea that causal diamonds in the boundary CFT are a useful way to charecterize boundary subregions in AdS/CFT has received attention since the works of [13, 14] which inaugurated the idea of *kinematic space*. In flat space, since the boundary has null components, a causal structure defined instrinsically on the boundary is ill-defined, so this idea does not work directly. But we will see that *bulk* causal diamonds attached to the conformal boundary can still be used to capture precisely analogous information. In particular, the fact that the space of certain pairs of points on the boundary remains of significance even in flat space, we find quite striking. Our work is also closely related to the HRT construction [12]: the choice of a point each on the two null boundaries defines a canonical class of HRT surfaces for flat space with a cut-off.

---

[7]This idea has its origins (I think) in covariant versions of entropy bounds, made precise by using light-sheets as the holographic screens by Bousso [8,9]. It turns out that this entropy is bounded by appropriately defined areas: this is the (Fischler-Susskind-)Bousso bound [8, 10, 11]. There is evidence [8] that the origin of these bounds is statistical and not thermodynamical, which again is a strong suggestion that it is a count on the number of degrees of freedom of a holographic fundamental theory. These ideas are one of the major sources for our work.

[8]See the appendix for a discussion on the precise role that an asymptotic causal diaomond plays in bulk reconstruction.

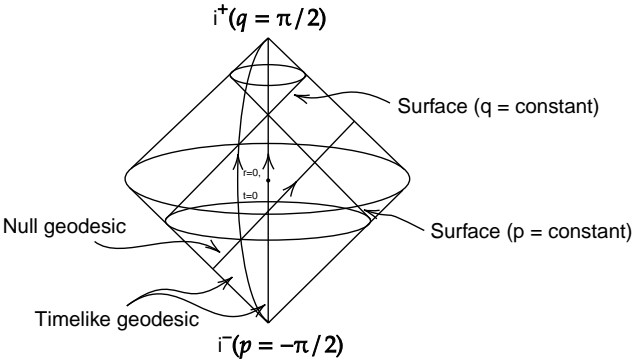

Figure 1: A plot of the $(t', r', \phi)$ coordinates of 3-dimensional flat space. The $r'$ coordinate is plotted radially, but it should be understood that the metric in the radial direction is not flat becaause of the conformal transformation. All points on the circle drawn at $r' = \pi$ are identified, and becomes the point at infinity. Our figure should be compared to figure 15 (i) in Hawking&Ellis. But we believe that what is labeled as a spacelike *geodesic* in the figure there is only a spacelike *curve*, at least for the 2+1 dimensional case that we consider here. We will have more to say about spacelike geodesics in this paper.

## 2 The Conformal Boundary of Minkowski Space

To set the stage, we will consider $d + 1$ dimensional Minkowski space, $M_{d+1}$, with $d = 2$ for concreteness and ease in drawing pictures – but we emphasize that the statements we make here generalize quite readily to all $d$. In many places, all one has to do is replace straight lines by hyperplanes and circles by (hyper)spheres.

Let us write down some formulas for the conformal structure of flat space to set up our notation: we largely follow the conventions of Hawking&Ellis. Flat space metric in polar coordinates $ds^2 = -dt^2 + dr^2 + r^2 d\phi^2$ is conformal to the Einstein static form

$$ds^2 = -dt^2 + dr'^2 + \sin^2 r' d\phi^2 , \qquad (1)$$

which is locally of the form $\mathbb{R} \times S^2$, but with a constrained range for $t', r'$:

$$-\pi < t' + r' < \pi, \ -\pi < t' - r' < \pi, \ r' \geq 0 . \qquad (2)$$

This metric is conformally flat, and upto the conformal factor (which will not be important for us) it turns into flat space under the coordinate change

$$t + r = \tan\left(\frac{t' + r'}{2}\right), \ \ t - r = \tan\left(\frac{t' - r'}{2}\right) . \qquad (3)$$

It is also convenient to introduce the null coordinates $p, q$ via

$$p + q = t', \ \ p - q = r' . \qquad (4)$$

In figure 1 we have drawn the Einstein static universe after opening up the $r'$ coordinates into a disc with an identified boundary circle. This is useful for maintaining the usual intuition for a radial coordinate, while working with the conformally compactified coordinates.

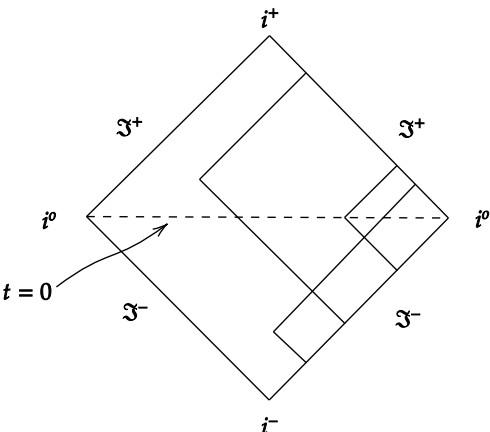

Figure 2: The basic idea of an asymptotic casual diamond, illustrated. We draw the 1+1 dimensional prototype here. If one views the *x*-coordinate as a radial coordinate together with a $\mathbb{Z}_2$ freedom (the angle degree of freedom in 1+1 d), this will turn into a more conventional Penrose diagram. The local bulk aspects we discuss are invisible in this simple 1+1 dimensional situation. We show only ACDs attached to the right boundary here.

## 3    Bulk Causal Diamonds Anchored to the Conformal Boundary

**Asymptotic Causal Diamonds:** Our goal is to construct an analogue of Rindler-AdS/entanglement wedge reconstrcution, that is useful in flat space. For this we find it useful to introduce the notion of an asymptotic causal diamond. The basic idea is that an asymptotic causal diamond in $M_{2+1}$ is defined by two points: one on the future boundary null cone $\mathcal{I}^+$ which we call $p_F$, and one on the past boundary null cone $\mathcal{I}^-$, which we call $p_P$. Note that the null boundaries $\mathcal{I}^\pm$ are defined by $p = \pi/2$ and $q = -\pi/2$ respectively, and therefore these points are fixed uniquely by one null coordinate (and the angles):

$$p_F = (\pi/2 - \epsilon, q, \phi), \quad p_P = (p, -\pi/2 + \epsilon', \phi') . \tag{5}$$

These points we will call the vertices of the diamond. We work with the conformal coordinates here to locate our asymptotic causal diamonds. We have exhibited the possibility of regulating the points at the boundary by introducing an $\epsilon, \epsilon'$ for convenience in some calculations when taking the asymptotic limit, but they can be set to zero[9]. The intersection of the past light cone of $p_F$ and the future light cone of $p_P$ defines an asymptotic causal diamond.

**Spacelike Geodesics:** Note that if we consider the future and past points to be ordinary points (instead of points at the conformal boundary) these intersections are simply circles [13, 15]. The difference, when we take these points to the conformal boundary is that these circles end up having infinite radius, and end up becoming (spacelike) straight lines. It is straightforward to show this systematically and we will present it soon, but the result is intuitive enough. This means that bulk regions that can be reconstructed from boundary data on the asymptotic causal diamond, are regions anchored at the boundary and bounded by straight lines[10]. The key point is that even though we have not specified an explicit bulk

---

[9]We view the space of asymptotic casual diamonds as part of the associated data of asymptotically flat space. Later we will also introduce a radial cut-off for some purposes. It will be interesting to relate this and the $\epsilon, \epsilon'$ more concretely.

[10]Let us emphasize a trivial point: the straight lines that we talk about here are straight lines in the original

reconstruction map[11] as in the AdS-Rindler wedge [17, 18, 21], we expect just from the fact it is the interior of an asymptotic causal diamond, that this entire region is re-constructible from the conformal boundary. This is the key requirement in building the connection with quantum error correction.

To present some explicit formulas, we can work with the spatial slice of the bulk that corresponds to the $t = 0$ instant (which is identical to the $t' = 0$ instant) as is usually done in the AdS/CFT case in discussions of bulk reconstruction and code subspaces. This slice has a simple description in terms of asymptotic causal diamonds: simply make sure that we only consider those ACDs in (5) with

$$p = -q \equiv q_0, \ \phi = \phi' \equiv \phi_0 \ , \tag{6}$$

where $q_0, \phi_0$ are arbitrary (within their allowed ranges). In other words, these are the symmetric asymptotic causal diamonds. For this class, we can write down explicit formulas which are not at all ugly, and that is another motivation to make things explicit. By translating these coordinates back to the standard flat space coordinates, we can see that they are the limiting cases of the classes of casual diamonds defined by the two points[12]

$$(t, r, \phi) = (\pm \tilde{T}, \tilde{R}, \phi_0) \tag{7}$$

in the limit where $\tilde{T}, \tilde{R}$ are both going to infinity, but $\tilde{T} - \tilde{R}$ is held fixed (and is controlled by $q_0$). Explicit formulas are simple to write down, we will present one: the circle that is at the waist of the casual diamond turns into the promised straight line in this limit:

$$2r \cos(\phi - \phi_0) = \tilde{R} - \tilde{T} \ . \tag{8}$$

We will sometimes use $k \equiv (\tilde{R} - \tilde{T})/2$ in what follows for brevity. The asymptotic causal diamonds whose waists are defined by these straight lines are a natural generalization of the usual Rindler wedge.

**Metric on the Generalized Rindler Wedge:** We can define a Rindler-like metric on these asymptotic causal diamonds. Specializing (without any real loss of generality) to the case $\phi_0 = 0$ in (8) the following coordinate transformation brings the original flat space coordinates $(r, t, \phi)$ to a natural Rindler-like form in the variables $(R, T, \Phi)$:

$$R = \sqrt{(r - k/\cos\phi)^2 - t^2}, \ T = \tanh^{-1}\left(\frac{t}{r - k/\cos\phi}\right), \ \Phi = \phi \ . \tag{9}$$

The generalized Rindler form of the metric follows from this – the explicit form is straightforward, but of not much use here.

To see that this is a natural generalization of Rindler, note that when $q_0 = 0 = k$, the asymptotic causal diamond has vertices that are "half-way" to time-like infinity, and corresponds to the usual Rindler wedge. Then the "waist" of the causal diamond is a straight line that passes through the origin of the Minkowski space: $r \cos\phi = 0$. (We have again taken $\phi_0 = 0$ without loss of generality.) It is immediate to check in this case that the coordinate transformation (9) above reduces to the simple metric

$$ds^2 = -R^2 dT^2 + dR^2 + R^2 \cosh^2 T d\Phi^2 \ , \tag{10}$$

---

flat space. They can be shown to be circles on the $S^2$ of the conformally flat patch of the Einstein static universe corresponding to Minkowski space, see Appendix.

[11]Indeed, the nature of the precise holographic data and the form of the bulk reconstruction map in flat space must differ from the familiar ones in AdS/CFT. We will comment about this briefly in an Appendix.

[12]We can write explicit finite expressions for $\tilde{T}, \tilde{R}$ if we retain the $\epsilon, \epsilon'$.

when $k = 0$. This metric is the standard (spherical) Rindler metric. See [24] to see a recent discussion of a related but different generalization of the spherical Rindler metric.

From the perspective of causal structure and bulk reconstruction, what we have obtained via the ACD coordinates $(R, T, \Phi)$ above is a coordinate system that is similar to the Minkowski coordinates $(r, t, \phi)$. The former covers all of the (asymptotic) causal diamond in one chart, while the latter covers all of Minkowski space. In the Penrose diagram, the constant $R$ and $T$ slices have a structure in the ACD that has some similarities to the $r$ and $t$ coordinates in the full Minkowski space. A field theory at $R = R_{cut}$ on the ACD[13] could be a useful way to describe holographic data at the boundary of the ACD for bulk recnstruction. This is analogous to how $r = r_{cut}$ in standard Minkowski space[14] could be a useful way to capture holographic data in Minkowski space. Note in particular that the $R_{cut} \to \infty$ limit takes one to the conformal boundary of the ACD[15].

It is worth mentioning however that despite some parallels with AdS-Rindler and despite its apparent simplicity, the $(R, T, \Phi)$ coordinates have one important difference when it comes to bulk reconstruction: in higher than 1+1 dimensions, the metric on the ACD is[16] time-dependent. This means that the problem of solving the wave equations and identifying appropriate spacelike Green functions could be harder in an ACD than it was in Rindler-AdS. Nonetheless, as far as making our points are considered, this is merely a technicality: the causal structure is our key concern here. Let us also note that it is perhaps significant that at the cut-off, the geometry is de Sitter as we mentioned in a footnote.

**ACD Reconstruction:** Even thought we will not write an explicit ACD reconstruction map in this paper, the above discussions lead to a direct parallel with the causal wedge reconstruction picture in AdS, because we expect from the causal structure that the data at the boundary of the causal diamond (ie., the intersection of the boundary of ACD and the conformal boundary of Minkowski space) is enough to fix bulk data in the interior of the ACD.

From this point of view, the Rindler-AdS/causal wedge reconstruction and ACD reconstruction are parallel structurally as ingredients for local bulk reconstruction. We will use this fact to make the connection with code subspaces momentarily.

Before proceeding, let us also note that for any bulk point in Minkowski space, we can define the regions on the boundary that are spacelike separated from it. This set of points is a natural candidate for the region in the boundary from which we expect to have an analogue of *global* bulk reconstruction. Thanks to the previously noted fact that a radial cut-off in Minkowski space is natrually comaptible asymptotically with the conformal boundary, we hope to report on some progress on an explicit construction of this type in the near future [19].

**Geometry of Asymptotic Causal Diamonds in the Conformal Diagram:** It is useful for the ensuing discussion to have an understanding of the detailed geometry of ACDs. In the conformal coordinates on the Einstein static sphere, the straight lines (8) take the form

$$2 \tan(r'/2) \cos(\phi - \phi_0) = \tan q_0 \equiv 2k \ . \tag{11}$$

Note that $\tan r' \cos(\phi - \phi_0) = \text{const.}$ corresponds to great circles (geodesics) on the sphere[17],

---

[13]This is a field theory living on de Sitter space, which is believed to be well-defined if one takes the Bunch-Davies vacuum to define correlators [25]. Note that the spherical Rindler metric is a time-dependent background in $T$.

[14]This is a field theory living on the Einstein static universe. Such theories are well-defined, the scalar case is standard [26], fermionic and superysmmetric examples go back to the work of D. Sen [27,28].

[15]In a previous version of this paper, there were statements in this paragraph to the effect that the generalized Rindler wedge coordinates do not reach the boundary in a suitable way, and therefore cannot be used for reconstruction. This made it seem that one needed alternate coordinates on the ACD for bulk reconstructiom (and it was not clear what was the natural choice). This problem goes away now, because the premise was in error. I thank Vyshnav Mohan for bringing this to my attention.

[16]Note that since this is just flat space in some other coordinates, the isometries still contain Minkowski time translation. The point however is that in the natural coordinates here, the time direction $T$ is not an isometry.

[17]Remember that $r'$ is the polar angle on the Einstein static sphere.

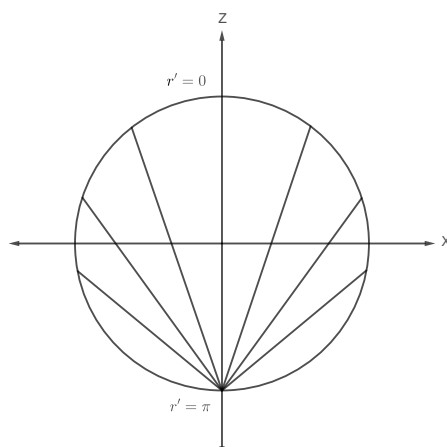

Figure 3: The $y = 0$ slice of the Einstein static sphere (which corresponds to $\phi_0 = 0$) where $x, y, z$ are auxiliary embedding coordinates. The poles are at $r' = 0$ (North) and $\pi$ (South). South pole is the point at infinity of $r$ in $M_{2+1}$. Changing $\phi_0$ corresponds to rotating this configuration around the $z$-axis.

and so these curves are *not* geodesics (naturally). But noting the similarity of the two forms suggests that the easiest way to have an intuition for these curves is to go to the Cartesian coordinates in which the sphere is embedded. When $\phi_0 = 0$, we find that these curves are the intersections of the planes

$$1 + z = \frac{x}{k} \, , \tag{12}$$

with the sphere $x^2 + y^2 + z^2 = 1$. These are circles, but not great circles. For different values of $k$, the projections of these circles on the $x$-$z$ plane are plotted in Figure 3.

In Figure 1, the Einstein static sphere is portrayed as a disc with its boundary points identified, and on the $t = 0 = t'$ plane, the above circles will be represented by the plots of (11). These curves connect diametrically opposite points: this is natural because a straight line in space provides two opposite ways to reach the point at infinity, and the fact that this line is (a finite distance) away from the origin becomes irrelevant at infinity. The schematic cross section on the equatorial plane, of three different causal diamonds is presented in Figure 4. We also present the picture of the ACD on a $(t', r', \phi)$ diagram where $r'$ is drawn as a radial coordinate, in Figure 5.

**Finite IR Cut-off and Code Subspace Structure:** With these, we have almost everything we need to show that the bulk local structure that arises out of reconstruction from boundary subregions in flat space is parallel to that in AdS. The key observation of [21] is that the bulk regions that are re-constructed from boundary subregions satisfy certain union and intersection rules, corresponding to the fact that they arise from unions and intersections at the boundary.

It can be seen that an identical structure arises here as well, if we use asymptotic causal diamonds to define bulk reconstruction together with a fixed (but arbitrary) radial cut-off. We will take this radial cut-off to be $r = r_{cutoff}$ or equivalently $r' = r'_{cutoff}$, but we do not expect the picture to change substantively for large classes of cut-offs. As is evident from the figures, the shape of the bulk reconstruction region has changed in AdS compared to flat space. In flat space they are straight lines as we emphasized before. But as long as the cut-off is finite, the union/intersection structure remains intact. This is entirely analogous to the situation in AdS.

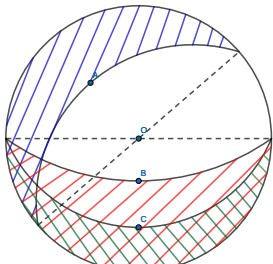

Figure 4: The "width" of the asymptoctic causal diamond is determined by how close to the equator its vertex is. A vertex at timelike infinity covers the entire Minkowski diamond, while a vertex "half-way" leads to the usual spherical Rindler wedge. The end points are determined by the anglular coordinates of the vertex and are independent of the vertex height of the ACD. They are diametrically opposite to each other, when there is no radial cut off. The curves are schematic.

Note that the unions and intersections are trivial to construct. Examples of the various kinds of configurations discussed in [21] are shown in figures. See our Figure 6 which should be compared to figure 3 in [21]. The causal reconstructibility of the bulk point from the conformal boundary is identical in both cases.

The discussion of the union/intersection structure is clear in standard polar coordinates with a cut-off, because we have regions bounded by straight lines and their unions and intersections are intuitive. But it is instructive to also consider the analogous picture in the conformal Einstein static coordinates: after all, we found in a previous discussion that the ACDs there span the same boundary half-circle irrespecive of how deep into the bulk they go, as long as the angular coordinates of their vertices are the same. This is clearly distinct from the AdS situation.

But again when we have an IR cut-off, the situation changes dramatically, and is illustrated in Figure 7. We find that indeed the union/intersection structure is identical to that which was found in [21]. Note also that as the tips of the causal diamonds move up(down) towards the future(past) timelike infinity, the causal diamond sweeps out the bulk region precisely once. In particular, this means that the spherical Rindler wedge – the case when the vertex is half-way up(down) in the conformal diagram – covers half of the boundary. This is consistent with the requirements of [21] (see eg., eqn. (4.28) in [29]) which requires half of the boundary for reconstructing the center of the bulk. Together, these facts show that the protection against boundary erasures follows a structure isomorphic to that in AdS.

One can also think about the above construction in terms of the causal structure of the cut-off geometry. For a radial cut-off, this is an Einstein static universe in one lower dimension ($\mathbb{R} \times S^1$ in the present case). We expect the HRT construction of the entanglement wedge of subregions on this boundary to lead to the same conclusions. What ACDs do is to provide a class of sub-regions that are natural from the perspective of the underlying casual structure.

**Scrambled Subregions and What Makes Flat Space Different:** There is one substantive way in which the above picture differs from AdS, however. The key point is that unlike in AdS, the sub-regions we have identified do not have a simple limit when the cut-off is taken to infinity. This should be clear from Figure 8. In AdS, the subregions of the conformal boundary when the cut-off has been taken to infinity give us a God-given set of boundary subregions.

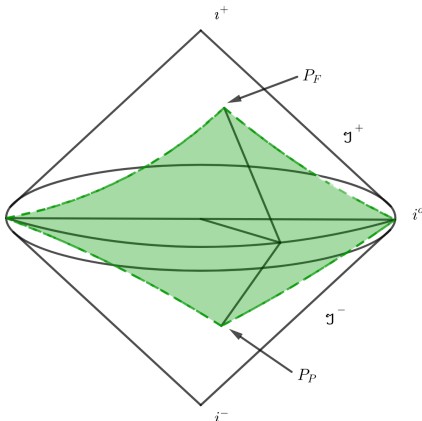

Figure 5: The shaded region is the "back side" of the asymptotic causal diamond, the region on the Minkowski diamond bounded by the boundary of this region is the "front" which we have not marked as to not clutter the figure. The inner curve connecting diametrically opposite points is a straight line (spacelike geodesic) in Minkoswki space. The region between this curve and the "lower"semi-circle is the interior of the ACD. As the vertices of the ACD move closer to the equator of the conformal diagram, the inner curve gets closer to the boundary. The curves in the figure are schematic, their connectivity is what we wish to emphasize.

This is related to the fact that gravity has decoupled at the boundary and therefore there is a canonical notion of sub-regions/tensor factors. Here we do not have such a simple limit. This is related to the fact we previously noted, that the ACDs (and therefore holographic data) spreads out all over half of the celestial sphere when the cut-off is taken to infinity. We expect that this is related to the non-locality of the dual theory and to the fact that the holographic entanglement entropy in flat space scales with the volume.

**Integral Geometry and Kinematic Space:** Let us take a moment here to note that the spacelike straight lines that bound our ACDs should be compared to the space of spacelike bulk geodesics that were used to define the CFT kinematic space in [14]. The works of [13–15] and various follow-ups, developed this notion of a CFT kinematic space, which in the AdS/CFT context can be viewed as the space of causal diamonds in the Minkowski space where the CFT lives. In flat space, since the conformal boundary does not have a well-defined causal structure, this perspective needs change. Interestingly, our proposal gives a natural bulk construction of the boundary kinematic space, for holography in flat space. It is noteworthy that the space of (suitably chosen) pairs of points seems to have a natural role in flat space as well.

Adapting results of [13,15], one can write down the metric on the space of our asymptotic causal diamonds, and potentially connect the wave equation on it, with the Casimirs of the Poincare group. This is clearly a direction that has substantial potential for development, especially in light of the fact that Poincare is a contraction of the conformal group. We will explore this in more detail in [20]. It will be interesting to see if we can define some version

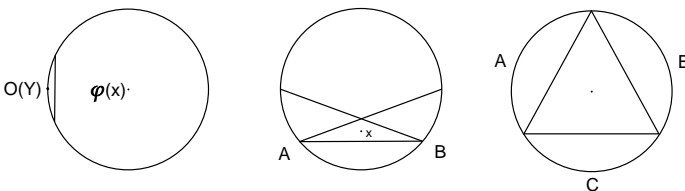

Figure 6: This figure should be compared to figure 3 in [21].

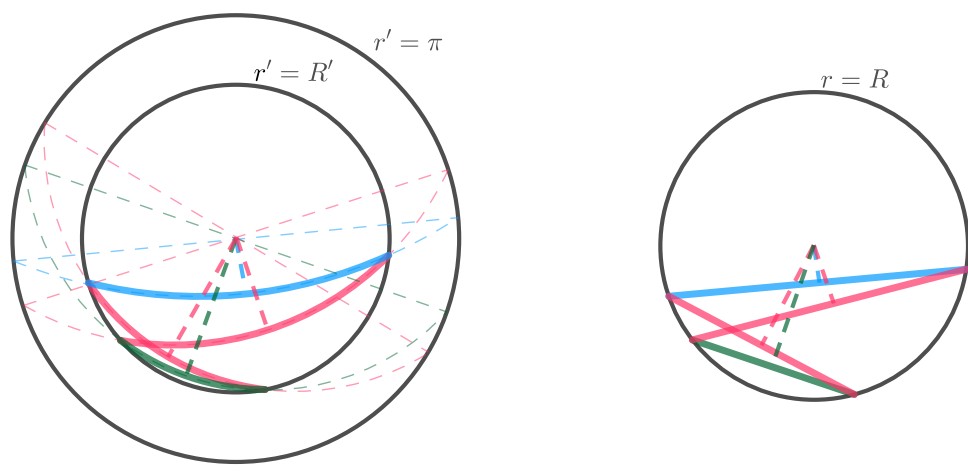

Figure 7: Comparison of the conformal coordinates and the standard Minkowski polar coordinates. The $r = \infty$ of polar coordinates gets mapped to $r' = \pi$. The union/intersection structures within the cut-off are isomorphic to that in AdS.

of a Poincare OPE block in a way analogous to the (conformal) OPE block defined in [15]. We expect also that the Kirillov-Kostant form on the co-adjoint orbit of the Poincare group should be related to the Crofton form on our kinematic space, in analogy with the AdS results of [30].

**Holographic Entanglement Entropy:** Let us conclude this section by noting that again at finite cut-off the spacelike lines (or hyperplanes in higher dimensions) match the previously noted Ryu-Takayanagi surfaces of flat space: see [32] for a Euclidean discussion and section 7 of [33] for a Lorentzian discussion which is closer in spirit to ours. Since these are minimal surfaces, the proofs of strong sub-additivity [34, 35] goes through as it does in AdS [36]. In fact in our 2+1 dimensional bulk, both the strong subadditivity statements regarding boundary sub-regions $S_{A+B} + S_{B+C} \geq S_{A+B+C} + S_B$ and $S_{A+B} + S_{B+C} \geq S_A + S_C$ turn into the statement that the sum of diagonals of a cyclic quadrilateral is greater than the sum of any two opposite sides: an immediate consequence of triangle inequality. It might be interesting to investigate other entangelement entropy inequalities in this setting. We will not pursue this line further since we do not have a separate definition of the dual theory, and have nothing to compare with. Note also that the holographic entanglement entropy here scales with the volume (of the cut-

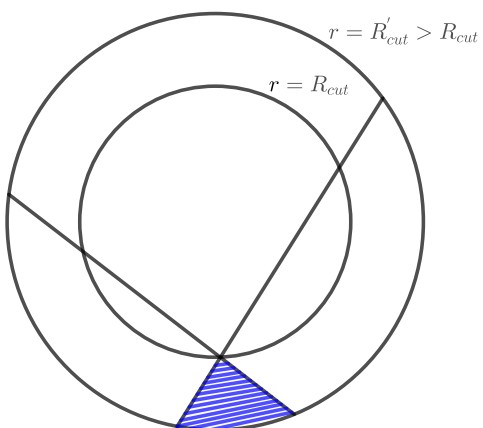

Figure 8: Sliding the cut-off outward demonstrates that scrambling never stops in flat space.

off) as it should, since the holographic dual of flat space is expected to have some non-local features. We will briefly comment on this again in the next section.

## 4 Comments

This paper is fairly broad in scope, and many directions of exploration automatically present themselves. We will only comment on a few limited aspects in this section.

In some of our discussions above, an IR cut off[18] played an interesting role. The precise nature of this cut-off was not too important for us (in the sense that we expect large classes of cut-offs to result in qualitatively identical conclusions). But here we will discuss holography in flat space, where the holographic screen is the wall of the box. Note that in this approach, the S-matrix (the usual observable in flat space) is to be viewed as a type of correlator where the IR cut-off has been taken to infinity in a suitable way at the end of the calculation[19]. We feel this perspective can be instructive, even if only quantities which have well-defined limits as the cut-off is tken to infintiy are ultimately what we are after.

To calculate holographic correlators in AdS, a finite IR cut-off has been quite useful from the beginning days of AdS/CFT (see eg., [42]). This is in spite of the fact that it is only in the strict $z \to 0$ limit of the AdS metric, that the AdS isometries reduce to conformal isometries at the boundary. At finite values of the cut-off, the bulk isometries move the cut-off[20], just as they

---

[18]Let us emphasize that it is the bulk IR cut off that we are referring to.

[19]Most investigations in flat space (eg., on soft theorems and BMS invariance) restrict their attention even further, to massless particles and the null boundary. See discussions and references in [37–41] for a point of entry into various perspectives. There are many interesting and clearly important ideas here, but we do not think a framework based entirely on the null boundary is a candidate for a *complete* description of holography in flat space, which should include massive particles.

[20]This is perhaps best viewed as a manifestation of the inexact decoupling between closed and open string modes as one moves away from the strict $z \to 0$ limit. A related observation is that a fully satisfactory holographic

do in flat space. An IR box for flat space has many parallels also to the discussions that arise in the context of AdS scattering amplitudes [45, 46]. There it is known that certain boundary correlators in the AdS box have a natural flat space S-matrix like limit. Even more strikingly, there is evidence there for the emergence of analogues of soft theorems [47]. We will view this as a suggestion that the soft-theorem/BMS/null-infinity aspects of flat space should be viewed as a sub-sector of the full set of holographic correlators that one can define in flat spacetime with an IR cut-off. Yet another indication that an IR cut off is worth exploring comes from [49], where fairly detailed evidence was presented that the thermal aspects of AdS black holes have natural analogues for flat space with box boundary conditions. Our discussions in this paper extend these previous considerations and presents evidence that parallel structures exist also for causal/local aspects of the reconstructed bulk in flat space.

To summarize, even if one's ultimate goal is the limit where the "cut off has gone to infinity", it might be useful to study boundary correlation functions for flat space with a finite cut-off. We will view this as a way to describe an *isolated* gravitating region in a holography-compatible way.

We will have more to say about the radial cut-off in future work [19], but for our present purposes, we have used $r = R_{cut}$ in standard spherical polar coordinates[21]. Note in particular that in the Penrose diagram of flat space, the $r$-coordinate in the $(t, r)$ coordinate grid has the nice property that the $r \to \infty$ limit takes us to the entire conformal boundary of the full Minkowski causal diamond. This suggests that a natural box where the holographic dual of flat space can live is on the Einstein static universe at $r = R_{cut}$ with coordinates $(t, \Omega_{d-1})$ where $\Omega_{d-1}$ stands for the angular coordinates of $d + 1$ dimensional asymptotically flat space. It will be interesting to calculate the boundary correlators on this space from the bulk at finite cut-off following adaptations of the standard GKP-W prescription [4, 5], and then see what interesting information can be extracted when taking suitable $R_{cut} \to \infty$ limits. We expect that a suitably defined set of *boundary* correlators of this type (or perhaps suitably identified S-matrix elements) will exhibit the full Poincare invariance of the *bulk*. Note also that the intersection of the asymptotic causal diamond with the cut-off surface offers a natural notion of causal structure on the cut-off surface[22].

One potential upshot of this discussion is to study field theory on the Einstein static universe [26–28] that gets suitably "frozen" as the radius of the sphere becomes infinite, and whose appropriate correlators attain an enhanced *bulk* Poincare invariance in this limit. S-matrix elements might be related to such correlators. It will be interesting to connect this with the discussions in [50, 51] which note that there is entanglement on the celestial sphere in flat space. It has been suggested that these theories have non-propagating local degrees of freedom, but non-local constraints and non-trivial entanglement. See, [52, 53]. This is also related to our previous discussions of holographic entanglement entropy in flat space.

In flat space, with only marginal extra baggage we were able to reproduce many of the structures that arose in AdS. Even though the ACD perspective has not been emphasized in AdS (because the kinematic space has many equivalent definitions there), one takeaway of our work is that it works equally well in both flat space and AdS. Let us make one comment about what is it that asymptotic causal diamonds capture. In AdS they directly capture a boundary subregion, which one can think of as a tensor factor of the Hilbert space of quantum gravity. A causal diamond encodes this tensor factor in a covariant way. In flat space, similar statements

---

renormalization group has been difficult to formulate in AdS/CFT, see [43, 44] for interesting attempts to make the scale-radius duality precise.

[21]Another natural candidate is to use an Ashtekar-Hansen [48] type radial coordinate.

[22]One of the problems with the null boundary is that it has no natural causal structure. Note however the trivial (but possibly useful) fact that not all curves on a null hypersurface are null. Eg: take a light cone, cut it by a $t =$constant surface. The resulting curve on the light cone is a spacelike circle.

might hold: an ACD gives us an abstract and covariant way to capture[23] the degrees of freedom of quantum gravity. This begs the question: what about other spacetimes? See [54–58] for some recent discussions about various aspects of boundaries in holographic settings.

Interestingly, there exists a perfectly well-defined notion of an asymptotic causal diamond in de Sitter space as well: they are defined by a point each, on the future and past boundaries. Typically one tries to view de Sitter holography in terms of its (spacelike) future/past infinity, it will interesting to see whether the present perspective is of some use as well. Note that clearly the structure of the space of ACDs has changed: unlike in AdS and (as we suggest in this paper) flat space, the ACDs are no longer straddling the boundaries. It remains to be seen whether this is a virtue or vice. A second interesting (and in many ways much simpler) example is the possibility of attaching "asymptotic" causal diamonds at the horizon of a black hole. In this picture, we view the outside region of the Kruskal diagram as a causally complete spacetime. This gives an approach for bulk reconstruction from the black hole horizon, as opposed to the boundary. The analogue of the IR cut-off we used here will be the stretched horizon. Note also that the structure of the union/intersections of these causal diamonds with those anchored at the boundary leads to some new ingredients to the code subspace picture here. There are a few concrete ideas to pursue here, we hope to report on some of them [59]. Note also that more general notions of ACDs are also possible in black hole spacetimes: in loose analogy with dS future/past boundary, we might also define them from the future or past singularities in black hole spacetimes. Similar ideas have appeared before, see eg. [60].

One of the key points we have emphasized here is that of causal structure. In other words, we are working with real time holography. Real-time AdS/CFT [61, 62] is not quite as well-developed as Euclidean AdS/CFT, but we expect that a picture analogous to [63] will work also in flat space, when we foliate the spacetime with the standard $r$ and $t$ coordinates in the Penrose diagram. It will be very interesting to develop this fully. Another direction that naturally presents itself, since pairs of points play a distinguished role in our construction, is that of bi-local holography of Das and Jevicki [64, 65]. One thing we have not mentioned at all in this paper is the connection between causal diamonds and tensor networks. This is clearly an idea worth exploring. See eg. [68, 69].

## Acknowledgments

I thank Budhaditya Bhattacharjee, (especially) K. V. Pavan Kumar, Alok Laddha, Raghu Mahajan, Vyshnav Mohan, Aninda Sinha, Ronak Soni and Amandeep Singh for discussions. I am particularly indebted to Aman for creating the pdf versions of my hand-drawn figures. I also thank the usual suspects at TIFR for stimulating questions and comments during a talk based on this material.

## A   Non-Standard PDE Data and Holography

Let us comment on a few points regarding bulk reconstruction from the ACD[24]. We will only discuss the scalar field in a fixed background below, which is the usual context of the HKLL like bulk reconstruction. Dynamical gravity introduces extra subtleties, which are not crucial for our purposes here.

It is best to not mix up the following two questions:

---

[23]As the reader will notice, the rub lies in the word "capture". To make it precise will require new ideas.

[24]A comment from Sandip Trivedi has influenced some of my thoughts in this section. I thank Gautam Mandal, Shiraz Minwalla and Sandip Trivedi for discussions.

- Is it possible to determine the scalar field in (some region of) the bulk if we are given the field and its normal derivative (as required for a second order PDE), on some *time*-like surface (say, $r = R_{cut}$) near the boundary[25]? A closely related question is, if there is such a region, what characterizes it?

- If the answer to the above question is yes, what are the constraints on such "Cauchy" data for it to be suitable for describing holography?

In AdS, that the answer is affirmative to the first question is an implicit (and often unemphasized) message of HKLL[26]. The explicit message of HKLL is the answer to the second question: the two pieces of "Cauchy" data in AdS are to be taken as the non-normalizable mode which we are instructed to set to zero, and the normalizable mode which we are free to specify. Given this "Cauchy" data, HKLL gives us an explicit construction of the bulk scalar field in terms of this "Cauchy" data.

Our claim is that in both AdS and flat space, for data provided on (an appropriate timelike cut-off of) an asymptotic causal diamond, the answer is "yes" to the first question. We will not try to answer the second question in this paper. All we are concerned with is the question of reconstructibility, given two appropriate pieces of "Cauchy" data. What further constraints should those two pieces satisfy for capturing various aspects of holography, is a question we will come back to in future work.

Now, let us provide some circumstantial evidence for the above claim for reconstructibility from a timelike surface. This is a somewhat non-standard type of data for second order hyperbolic PDEs, and even though the problem feels fairly basic, we have not been able to find references that deal precisely with the type of problem we are after[27]. The usual Cauchy boundary conditions involve data on a spacelike slice.

Let us first consider an ordinary (ie., non-asymptotic) causal diamond in Minkowski space. Consider a timelike surface that goes through its vertices and also through the point at its center [28]. The intersection of this surface with the interior of the causal diamond, we will call a *timelike cross-section* of the causal diamond[29]. We claim that the "Cauchy" data, aka. the value of the field and its derivative on a timelike cross-section, is precisely enough data for reconstructing the field everywhere inside the causal diamond. In 1+1 dimensions this is straightforward to see, for massless scalar waves. We can simply use the fact that flipping $t \leftrightarrow r$ only results in an overall sign change of the wave equation and the fact that timelike slices are Cauchy surfaces for evolution along $r$. Another (more explicit) way to come to the same conclusion is to use d'Alembert's solution to the 1+1 dimensional wave equation (see for example eq. (6.41) in [66]) and express the "Cauchy" data in terms of the Cauchy data (ie., the functions $\phi_0$ and $v_0$ in [66]). It can be seen that this is a bijection. This is also related to the uniqueness properties of solutions of the wave equation in their domains. Generalizations of d'Alembert's solution exist in all dimensions. One can also relate the spacelike and timelike data by using Fourier series to deompose the solution.

We expect that versions of these arguments should hold also in higher dimenions, and also when there is mass [67]. One reason to suspect this is that the standard obstruction to

---

[25]We will call such data, "Cauchy" data with quotes, to differentiate it from the unquoted Cauchy data, which is given on spacelike slices.

[26]Note that a Rindler-AdS wedge associated to a (spherical) boundary subregion in which we can do this reconstruction is the natural notion of an asymptotic causal diamond of AdS.

[27]This does *not* necessarily mean that they do not exist. The trouble is partly that it is hard to find suitable keywords for google, that zero in precisely on what we are after, especially on a topic as old and vast as PDEs.

[28]Letting the slice pass through the center is merely a matter of convenience: we expect data on any timelike slice to work.

[29]Note the crucial fact that when a causal diamond goes to infintiy and becomes an ACD, the center of the diamond tends to the boundary! So if we can re-construct the diamond from its central time-like slice, it is natural to expect that ACDs can re-construct the bulk.

development of a hyperbolic PDE from a (non-null) surface for "Cauchy" data is the existence of "charatecteristic surfaces". These are basically places where the normal to the surface becomes zero norm, *aka* these are just null surfaces.

Based on these, we conjecture that in both AdS and flat space, the (suitably regulated) outer boundary of an asymptotic causal diamond (ACD) is a region where if you give "Cauchy" data, you can do evolution into the spacelike bulk, and the region in which you can do this reconstruction, is precisely what defines an ACD. This is of course, demonstrably true in AdS thanks to the Ridler-wedge version of the HKLL construction. It will be very interesting to do something similar explicitly in flat space [67].

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
