# Peer review of "Bulk Locality and Asymptotic Causal Diamonds"

_SciPost Physics, doi:SciPost Phys. 7, 057 (2019)_

## Round 1 · Referee Report · Anonymous · 2019-9-2

Report

In this manuscript, the author generalized the AdS-Rindler reconstruction of bulk operators to flat space by introducing the notion of an asymptotic causal diamond. The author observed that this could be a useful way of organizing quantum information in flat space holographically.

I have the following questions which I hope will help improve the clarity of the manuscript once they are addressed:

1. Is there an explicit reconstruction formula in an asymptotic causal diamond, in analogy with the HKLL formula in an AdS-Rindler wedge?

2. Is there a way to see the story work in a concrete example of flat space holography, such as the BFSS matrix model?

  • validity: -
  • significance: -
  • originality: -
  • clarity: -
  • formatting: -
  • grammar: -

Author:  Chethan Krishnan  on 2019-09-10  [id 599]

(in reply to Report 1 on 2019-09-02)
Category:
remark
answer to question
pointer to related literature

My (very incomplete) answers to the queries are below -

  1. Unfortunately, in dimensions higher than 1+1, the wave equation in our generalized Rindler wedge coordinates is not separable. But it is possible to argue that the reconstruction must exist, because this is essentially an existence problem for solutions of hyperbolic PDEs in regions where one does not have to cross charecteristic (ie., null) surfaces etc. A paper where we give a detailed analysis of the "holographic content" in this sense, of an asymptotic casual diamond in general dimensions, is currently in preparation. The basic idea is as outlined in the Appendix of the present paper.

2.  This is an interesting question that I have not been able to seriously consider yet. My first thoughts have been that perhaps one could relax the notion of the infinite momentum frame (IMF) used in the BFSS matrix model to an asymptotically infinite momentum frame (perhaps in the future and past). But while this would help connect with the asymptotic causal diamond idea, it is not clear to me that there is a manageable description of M-theory when one is away from the IMF. But it seems very likely (considering the crucial role played by the IMF in BFSS), that something like our asymptotic causal diamonds may play a role. One paper/review which I found to be thought-provoking for these types of considerations is Polchinski's hep-th/9903165, in particular section 4.

---

## Editorial Decision

published